# Reading Relevant Feature from Global Representation Memory for Visual Object Tracking

**Xinyu Zhou**[1]    **Pinxue Guo**[2]    **Lingyi Hong**[1]    **Jinglun Li**[2]    **Wei Zhang**[1]

**Weifeng Ge**[1*]   **Wenqiang Zhang**[1,2*]

[1]School of Computer Science, Fudan University, Shanghai, China
[2]Academy for Engineering and Technology, Fudan University, Shanghai, China
zhouxinyu20@fudan.edu.cn, pxguo21@m.fudan.edu.cn,lyhong22@m.fudan.edu.cn,
jingli960423@gmail.com, weizhang@fudan.edu.cn,
weifeng.ge.ic@gmail.com,wqzhang@fudan.edu.cn

## Abstract

Reference features from a template or historical frames are crucial for visual object tracking. Prior works utilize all features from a fixed template or memory for visual object tracking. However, due to the dynamic nature of videos, the required reference historical information for different search regions at different time steps is also inconsistent. Therefore, using all features in the template and memory can lead to redundancy and impair tracking performance. To alleviate this issue, we propose a novel tracking paradigm, consisting of a relevance attention mechanism and a global representation memory, which can adaptively assist the search region in selecting the most relevant historical information from reference features. Specifically, the proposed relevance attention mechanism in this work differs from previous approaches in that it can dynamically choose and build the optimal global representation memory for the current frame by accessing cross-frame information globally. Moreover, it can flexibly read the relevant historical information from the constructed memory to reduce redundancy and counteract the negative effects of harmful information. Extensive experiments validate the effectiveness of the proposed method, achieving competitive performance on five challenging datasets with 71 FPS.

## 1 Introduction

Visual object tracking (VOT) is a fundamental task in computer vision, which provides only the location of an object in the first frame and then accomplishes continuous predictions of the object's position in subsequent video frames. At present, visual object tracking is widely used in autonomous vehicle [25], video surveillance[48], and other scenes. As video frames are dynamic and continuously changing, it presents a significant challenge in addressing the appearance variation of the target and the changes in the environment. The current mainstream approach to addressing this problem is in two perspectives: the first type[47, 5, 54, 41, 32, 49, 26], as shown in Fig. 1(a), is to perform interactive computations on all patches of a fixed template and the search region. The second type[15, 51, 52, 9, 14, 43], as shown in Fig.1(b), is to perform the interactive computation of template memory and the search region. Despite their great success, they neglect the appearance variation of the target and the changes in the environment. In other words, the relevant reference information for the search region at different time steps should be diverse. Retrieving all information from the

---

*corresponding author.

37th Conference on Neural Information Processing Systems (NeurIPS 2023).

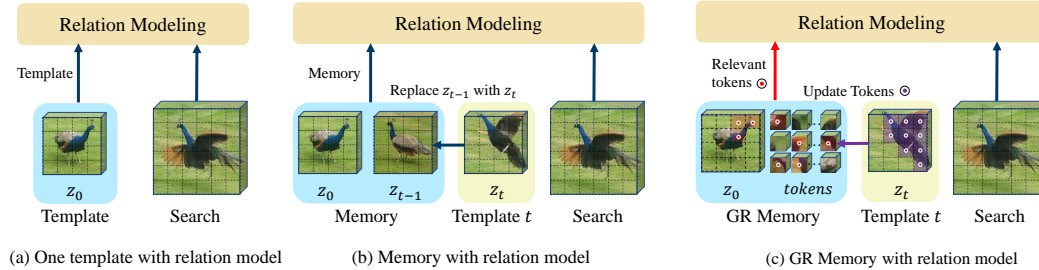

(a) One template with relation model   (b) Memory with relation model   (c) GR Memory with relation model

Figure 1: Three different methods of tracking pipeline. The purple dots represent the selected points from the new template that are updated into the memory, while the red dots and blue dots represent the selected points that are fed into the relation model.

template or memory to the search region is redundant, and some unnecessary historical information that does not match the current time step may potentially degrade tracking performance.

In this paper, we propose a novel tracking paradigm (RFGM) to address the aforementioned problems. As shown in Fig.1(c), the core insight of our approach is to extract relevant historical cues from memory for different search frames at different time steps. The approach dynamically selects historical information from memory that is more suitable for the current search region, thereby improving tracking performance. Additionally, to obtain the target appearance features of the entire video sequence as reference features, we utilize the dynamic selection ability of our tracker to construct a global representation memory(GR Memory), which dynamically stores the target appearance features of historical frames for adapting to the appearance variation of the target and improving the robustness of the model.

Specifically, as shown in Fig.1(c) we design relevance attention specifically for tracking, which is different from previous attention[54, 51, 5] based on global information reading. The relevance attention mechanism can read global information across frames on the timeline of the video sequence, and decide which token to use in the attention mechanism based on dynamic ranking. The most suitable historical information for the current frame is adaptively selected from the memory through the attention mechanism. Each stage of the relevance attention discards unselected tokens, so the computational complexity of later layers is smaller than that of previous layers. Finally, to construct the global representation memory, unlike directly replacing the earliest template with a new one in Fig.2(b), we utilize the relevance attention to automatically select the target tokens from the new template and the original memory as the new memory. The constructed GR memory retains the most important target features of the original memory and updates new target features into the memory, as presented in Fig.1(c). After multiple updates, it will obtain the most representative target features in the entire video sequence.

Our contribution can be concluded in three-fold: (**i**)We propose a novel tracking framework that adapts to changes in target appearance and background by constructing a global representation memory at the token level across frames and reading from this memory to capture the most relevant features at the current time step;(**ii**) We design a relevance attention mechanism for the search region to selectively extract template features from memory. Simultaneously, it is utilized to update the global representation memory at the token level, reduce memory consumption and enhance tracking speed; (**iii**) We conduct systematic experiments and validate the effectiveness of the proposed designs. Our tracker achieves competitive performance on five widely used benchmarks.

## 2 Related work

### 2.1 Visual Object Tracking Paradigms

In the past few years, many trackers have achieved great success in the field of visual object tracking. Typically, the linear matching method with cross-correlation is used to locate the target position in the search frame[1, 49, 6, 42, 26, 16, 57, 44, 2]. SiamFC[1] is the first to propose the use of Siamese networks for feature extraction and linear correlation operation for visual object tracking.

SiamFC++[49] further extends this method by feeding the features after linear correlation into three different prediction heads for classification, quality assessment, and regression, respectively, to decompose the tracking task. After the emergence of the transformer, the cross-attention operation[15, 41, 9, 14, 43, 54, 5, 32, 47, 51] based on the attention mechanism replace the original cross-correlation as the mainstream paradigm for object tracking. TransT[5] incorporates a CFA module to achieve bidirectional interaction between the template and the search region. AiAtrack[15] utilizes an attention in attention mechanism to achieve matching from the template to the search region, while OSTrack[54] and MixFormaer[9] use a one-stream paradigm to extract features and interact between the reference features and search region simultaneously.

However, the historical information required for different search regions at different time steps should be different. Therefore, cross-correlation and cross-attention operations introduce redundant information, potentially deleting tracking performance. To address this problem, we propose a novel tracking paradigm, RFGM, which automatically extracts the most suitable information for the search region from the GR memory, improving the adaptability to the appearance variation of the target and the changes in the environment.

## 2.2 Attention mechanism

The attention mechanism is widely employed in the tracking domain, and it has demonstrated remarkable performance, as exemplified by TransT[5], STARK[51], and OSTrack[54], etc. However, these mechanisms integrate information from all templates with the information from the search region, resulting in redundant template features. In contrast, deformable attention possesses feature selection. Deformable attention, derived from deformable CNN[10, 59], has been applied in various fields such as object detection[3] and image classification[19]. It can adaptively select neighboring tokens based on the current query. For instance, Deformable DETR[60] uses deformable attention to regress the offset of the neighborhood based on the query coordinate in order to assist the query in selecting the most appropriate key and value, which accelerates the convergence of the DETR model for object detection. DAT[46] extends deformable attention to the backbone for classification. However, these deformable methods are based on regressing the coordinates of neighboring tokens within the current frame and cannot globally read all tokens across frames. Inspired by Dynamic ViT[38], we design a relevance attention mechanism specifically for visual object tracking. Instead of regressing the coordinate offsets, we utilize ranking to dynamically assist the search region in globally reading useful features across historical frames.

## 2.3 Memory networks

Information from historical frames enables the network to adapt to the variation of appearance and background. Many approaches are devoted to template updates for improving performance in different fields, such as video object segmentation[36, 17, 53, 8, 21, 7, 58], video obejct tracking[52, 14, 51, 9], etc. In video object segmentation, STM[36] and STCN[8] construct memory networks, enabling the model to store extensive historical frame information for robust object tracking and segmentation. However, since memory incurs memory overhead and reduces model computation speed, it becomes challenging to store all historical frames during long-term video object segmentation. Therefore, XMem[7] introduces three distinct types of memory to facilitate long-term object tracking and segmentation. In object tracking, GradNet[27] updates the template by backwarding the gradients. With a template controller based on LSTM, MemTrack [52] design a memory network for robust object tracking. STMTrack[14] proposes a memory-based tracker, which updates templates at a fixed interval. Besides, Mixformer[9] and STARK[51] design a scoring head to select the representative template. We utilize the proposed relevance attention to construct a GR memory. In contrast to previous methods that update the memory with the entire template, the GR memory adaptively selects relevant tokens from both the original memory and new template tokens using relevance attention. After multiple updates, the GR memory contains the most representative tokens of the target in the entire video sequence.

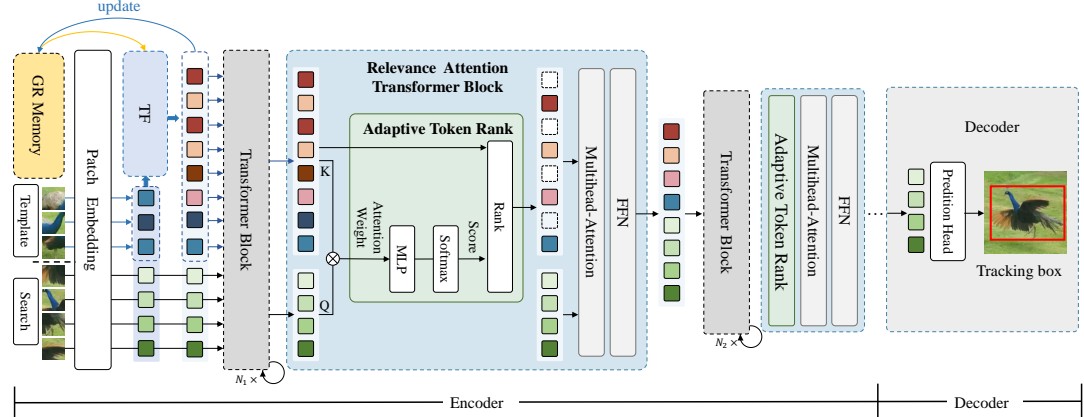

Figure 2: The framework of RFGM. It consists of a GR memory, token filter(TF), an encoder, and a decoder. The encoder is composed of Attention and relevance attention, while the decoder consists of a prediction head.

## 3 Method

### 3.1 Tracking with GR memory and relevance attention

**Pipeline**.

$Encoder$. As shown in Fig.2, The RFGM utilizes the vanilla ViT[12] as the encoder for feature interaction. The initial input of RFGM is a pair of images, which is referred to as template $z \in \mathbb{R}^{3 \times H_z \times W_z}$ and search $x \in \mathbb{R}^{3 \times H_x \times W_x}$. The template and search in RFGM are first divided into $N_z$ and $N_x$ non-overlapping patches $z^p$ and $x^p$, respectively. The size of these patches is $S \times S$, and $N_z = H_z W_z / S^2$, $N_x = H_x W_x / S^2$. These patches are then linearly mapped to a series of patch embeddings (tokens) by a convolutional layer, $T^z = \{T_1^z, T_2^z, \cdots, T_{N_z}^z\}$ and $T^x = \{T_1^x, T_2^x, \cdots, T_{N_x}^x\}$, $T \in \mathbb{R}^{1 \times C}$, where $C$ is the dimension of patch embedding. We concatenate the $T^x$ and initial $T^z$ and feed them into the encoder to obtain the tracking results of the first frame. In the subsequent tracking process, the predicted box will be used to crop a new template to adapt to changes in the target's appearance. Besides, we establish a GR memory $M$ to store the most representative target features throughout the video. Then, we feed $M$, a new template, and $T^x$ into the Token Filter (TF) module to get the selected tokens from memory and the new template. Finally, we feed the selected token and $T^x$ into the encoder for feature interaction. The entire encoder consists of 12 transformer blocks, with relevance attention transformer blocks employed at the 4-th, 7-th, and 10-th layers.

$Decoder$. The prediction head is based on the decoder of [54] and consists of three branches: score branch, offset branch, and size branch. Each branch is composed of three convolutional layers. The score branch is used to predict the position $R \in [0,1]^{\frac{H_x}{16} \times \frac{H_x}{16}}$ of the target center, the offset branch is used to compensate for the error $E \in [0,1]^{2 \times \frac{H_x}{16} \times \frac{H_x}{16}}$ caused by downsampling, and the size branch predicts the height and width $O \in [0,1]^{2 \times \frac{H_x}{16} \times \frac{H_x}{16}}$ of the target. The position with the highest score in the score map of the score branch is selected, $i.e.$, $(x_r, y_r) = argmax(x_r, y_r)R_{xy}$ and combined with the target size to obtain the final bounding box:

$$(x, y, w, h) = (x_r + E(0, x_r, y_r), y_r + E(1, x_r, y_r), O(0, x_r, y_r), O(1, x_r, y_r)) \qquad (1)$$

**Global Representation Memory**. Previous visual object tracking methods[15, 9, 43, 51] generally push a template $z_t$ to memory $M_{t-1} = \{z_0, z_1, \cdots, z_{t-1}\}, M \in \mathbb{R}^{n \times 3 \times H_z \times W_z}$ at fixed time intervals and discard the earliest template $z_1$ in a first-in-first-out manner, where $n$ is the number of templates, and $t$ is the time step in the video. $z_0$ is the template cropped with ground truth, it will be stored in memory all the time. The memory updating can be formulated as:

$$M_t = \{z_0, z_2 \cdots, z_{t-1}\} \cup z_t \qquad (2)$$

However, directly pushing the entire template to memory will introduce distractor information, and discarding the earliest template is prone to losing representative features.

Figure 3: The token filter consists of three regular transformer blocks and an adaptive token rank, which effectively updates the features in memory. GR-M represents the global representation memory, T stands for the new template, and S represents the search region.

To address the aforementioned issues, instead of updating the whole template, we propose a global representation memory (GR memory) updating as token level, which can adaptively select the most representative tokens from both the previous time step memory $M_{t-1}$ and new template of $z_t$. We initialize a memory $M_0 = \{T^{z_0}\}$ and add the new coming template tokens $T^{z_t}$ of $z_t$ to it when the memory is not full. When the memory $M_{t-1} = \{T^{z_0}, T^{z_1}, \cdots, T^{z_{t-1}}\}$ reaches a certain maximum number $N_{max}$ of tokens, as shown in Fig.3, we feed $M_{t-1}$ and $T^{z_t}$ into the Token Filter module to obtain relevant scores, and subsequently rank these tokens score in order, selecting the top $k$ tokens that are most relevant to the current search region and merge them with the initial memory $M_0$ to obtain the GR memory $M_t$. Because $T^p = \{T_1^z, T_2^z, \cdots, T_{N_z}^z\}$, the GR memory updating can be formulated as:

$$m = Topk(Rank(M_{t-1} \cup T^{z_t})) \tag{3}$$

$$\begin{aligned} Rank(M_{t-1} \cup T^{z_t}) = Rank(\{T_1^{z_1}, T_2^{z_1}, \cdots, T_{N_z}^{z_1}, T_1^{z_2}, T_2^{z_2}, \cdots, T_{N_z}^{z_2}, \cdots, T_1^{z_{t-1}}, T_2^{z_{t-1}}, \\ \cdots, T_{N_z}^{z_{t-1}}\} \cup \{T_1^{z_t}, T_2^{z_t}, \cdots, T_{N_z}^{z_t}\}) \end{aligned} \tag{4}$$

$$M_t = M_0 \cup m = \{T_1^{z_0}, T_2^{z_0}, \cdots, T_{N_z}^{z_0}, T_1^{z_1}, T_{10}^{z_2}, T_3^{z_5}, \cdots, T_{N_z}^{z_t}\} \tag{5}$$

where $m = \{T_1^{z_1}, T_{10}^{z_2}, T_3^{z_5}, \cdots, T_{N_z}^{z_t}\}$ is the selected top $k$ tokens, the size of $m$ is $N_{max}$. The way of memory updating can automatically discard some distractor information and select the most suitable tokens for the search region. Through multiple rounds of updates, the final memory includes the most representative target tokens throughout the entire video sequence.

**Relevance attention**. Previous feature interaction for the search region typically involves reading all information from a regular memory $M_t = \{z_0, z_1, \cdots, z_t\}$ using linear correlation[49, 1] or cross-attention[54, 9], which can be represented by the following formula:

$$x_* = M_t \star x = \sum_{i=0}^{t}(z_t \star x) \tag{6}$$

$$q = x, k = v = M_t, x_* = x + MHA(q, k, v) \tag{7}$$

where $\star$ is linear correlation[49, 1], the variables q, k, and v are used to denote the query, key, and value that are inputted into the multi-head attention($MHA$) block[12]. However, utilizing all the information in the memory is redundant and some harmful information may deteriorate the performance.

Therefore, we propose relevance attention that can automatically select the most relevant tokens from memory, which can be represented by the following formula:

$$m' = Topk(Rank(M_t^l)) \tag{8}$$

where $M_t^l$ is the memory tokens at time step t and serves as the input to the $l$-th layer of the transformer block. Then, we take the tokens $m'$ that are most relevant to the current search region as the query, key, and value. We also add the search region tokens $T_l^{x_t}$ to the query, key and value, resulting in the

final query, key, and value. The relevance attention can be formulated as:

$$q = k = v = m' \cup T_l^{x_t}$$
$$o^l = m' \cup T_l^{x_t} + MHA(q, k, v) \tag{9}$$
$$o_*^l = o^l + FFN(o^l)$$

where $T_l^{x_t}$ represents the search region tokens of $l$-th layer transformer block, $FFN$ represents a feed-forward network (FFN), and $o_*^l$ is the output of $l$-th layer transformer block.

## 3.2 Adaptive Token Ranking

To enable ranking in relevance attention and GR memory, we design an adaptive token ranking module in the 4-th, 7-th, and 10-th transformer layers. Attention weights can represent the relevance between tokens in memory and the current search area feature. Therefore, we input the attention weights into a multi-layer perceptron(MLP) to enhance the relevance, obtaining the relevant score for the current search area feature. Finally, we select the most relevant tokens by adaptive token ranking.

To simplify, we only use $T^{z_t}$ of one template from the memory to demonstrate the formula. Therefore, the multi-head attention weights of $T^{z_t}$ and $T^{x_t}$ in $l$-th layer in encoder is $w \in \mathbb{R}^{h \times N_z \times N_x}$, where $h = 12$ is the number of multi-head. The score prediction can be represented as follows:

$$w' = \frac{\sum_{j=0}^{N_x} w_j}{N_x}, w_j \in \mathbb{R}^{h \times N_z} \subseteq w \tag{10}$$

$$\pi = Softmax(MLP(w'^T)) \in \mathbb{R}^{N_z \times 2} \tag{11}$$

where $\pi_{y,0}$ represents the score of selecting the $y$-th token, $\pi_{y,1}$ represents the score of discarding the $y$-th token, $y \in [0 \sim N_z)$

**Training stage**. We keep all tokens that $\pi_{y,0}$ is greater than $\pi_{y,1}$. As the token selection based on the score is non-differentiable, we exploit the Gumbel-Softmax function[23] to enable gradient backpropagation. It can be formulated as:

$$D = Gumbel - Softmax(\pi) \in \{0, 1\}^{N_z} \tag{12}$$

where $D$ is a one-hot tensor with length $N_z$, the differentiability of Gumbel-Softmax enables end-to-end training. Finally, multiplying $D$ with attention weights $w$ to discard low-scoring tokens:

$$w_{masked} = Concat(w_{0,0}D, w_{0,1}D, \cdots, w_{m,n}D), m \in [0, h), n \in [0, N_x) \tag{13}$$

where $w_{masked}$ represents the masked attention weights, and $Concat$ represents recombining all $(w_{m,n}D \in \mathbb{R}^{N_z})$ to $w_{masked} \in \mathbb{R}^{h \times N_z \times N_x}$.

**Inference stage**.Instead of simply using $D$ as a mask for binary classification (keeping or discarding), we rank the scores of all tokens in $T^z = \{T_1^z, T_2^z, \cdots, T_{N_z}^z\}$ and select the top $k$ tokens, which allows selecting the tokens that are most relevant to $T^x$. It can be formulated as:

$$T_*^z = Topk(Rank(T^z)) \tag{14}$$

## 3.3 Loss Fucntion

In the training stage, we generate a Gaussian map using ground truth and use focal loss[29] to supervise the score branch. In addition, we use L1 loss and GIoU[39] loss to supervise the offset and size branches respectively. We also set up a ratio loss to supervise $D$ in the adaptive token ranking, which can help us to constrain the ratio of kept tokens:

$$L_{ratio} = \frac{1}{BS} \sum_{b=1}^{B} \sum_{s=1}^{S} (q^{(s)} - \frac{1}{N_z} \sum_{j=1}^{N_z} D_j^{b,s})^2 \tag{15}$$

where $q = [0.9, 0.8, 0.7]$ represents kept target ratio for $s$ stages, and stages include 4-th, 7-th, 10-th layer in encoder. $B$ is the batchsize. Finally, the total loss of a combination of the above objectives:

$$L_{total} = \lambda_{score}L_{focal} + \lambda_{iou}L_{giou} + \lambda_{l1}L_{l1} + \lambda_{ratio}L_{ratio} \tag{16}$$

where $\lambda_{score} = 1$, $\lambda_{iou} = 2$, $\lambda_{l1} = 5$ and $\lambda_{ratio} = 1$ are the weights to balance the objectives.

Table 1: Comparisons with state-of-the-art methods with search resolution $< 300 \times 300$ on three large-scale benchmarks. The top three metrics are highlighted with **red**, blue, and green fonts.

| | Method | TrackingNet [34] | | | GOT-10k* [22] | | | LaSOT [13] | | |
|---|---|---|---|---|---|---|---|---|---|---|
| | | AUC | $P_{Norm}$ | P | AO | $SR_{0.5}$ | $SR_{0.75}$ | AUC | $P_{Norm}$ | P |
| | **RFGM-B256** | **84.7** | **89.6** | **83.6** | **74.1** | **84.6** | **71.8** | 70.3 | **82.0** | **76.4** |
| Resolution $< 300 \times 300$ | SimTrack [4] | 83.4 | 87.4 | - | 69.8 | 78.8 | 66.0 | 70.5 | 79.7 | - |
| | OSTrack-256 [54] | 83.1 | 87.8 | 82.0 | 71.0 | 80.4 | 68.2 | 69.1 | 78.7 | 75.2 |
| | SwinTrack [28] | 81.1 | - | 78.4 | 71.3 | 81.9 | 64.5 | 67.2 | - | 70.8 |
| | SLT [24] | 82.8 | 87.5 | 81.4 | 67.5 | 76.5 | 60.3 | 66.8 | 75.5 | - |
| | SBT [47] | - | - | - | 70.4 | 80.8 | 64.7 | 66.7 | - | 71.1 |
| | AutoMatch [56] | 76.0 | - | 72.6 | 65.2 | 76.6 | 54.3 | 58.3 | - | 59.9 |
| | TransT [5] | 81.4 | 86.7 | 80.3 | 67.1 | 76.8 | 60.9 | 64.9 | 73.8 | 69.0 |
| | SiamAttn [55] | 75.2 | 81.7 | - | - | - | - | 56.0 | 64.8 | - |
| | SiamBAN [6] | - | - | - | - | - | - | 51.4 | 59.8 | - |
| | DSTrpn [40] | 64.9 | - | 58.9 | - | - | - | 43.4 | 54.4 | - |
| | Ocean [57] | - | - | - | 61.1 | 72.1 | 47.3 | 56.0 | 65.1 | 56.6 |
| | SiamPRN++ [26] | 73.3 | 80.0 | 69.4 | 51.7 | 61.6 | 32.5 | 49.6 | 56.9 | 49.1 |
| | MDNet [35] | 60.6 | 70.5 | 56.5 | 29.9 | 30.3 | 9.9 | 39.7 | 46.0 | 37.3 |

## 4 Experiments

### 4.1 Implementation Details

**Model**. The proposed RFGM uses ViT-B as the encoder, namely RFGM-B256. ViT-B is initialized using the pretrained weights from the MAE model[18]. RFGM-B256: the search region, which is $4$ times the area of the target object, is resized to $256 \times 256$ pixels. The template, which is 2 times the area of the target object, is resized to $128 \times 128$ pixels. The proposed relevance attention consists of three linear layers and the GELU activation function[20]. The output dimensions of the three linear layers are 384, 192, and 2, respectively.

**Training**.Our experiments are conducted on Intel(R) Xeon(R) Gold 6326 CPU @ 2.90GHz with 252GB RAM and 4 NVIDIA GeForce RTX 3090 GPUs with 24GB memory. This model training is divided into two stages. In the first stage, the number of templates is set to 3 frames and trained for 300 epochs, with 60k image pairs per epoch. The learning rate decreases by a factor of 10 after 240 epochs. In the second stage, fine-tuning is performed based on the first stage, with the number of templates increased to 7 frames and trained for 50 epochs. For GOT-10k, the model is trained for 100 epochs and decreases the learning rate at epoch 80 in the first stage. In the second stage, the model is finetuned for an extra 30 epochs. We set the learning rate of the prediction module in the relevance attention of the encoder and the decoder to 4e-4 and set the learning rate of the remaining parameters in the encoder to 4e-5. Additionally, the weight decay is set to 1e-4. Our training datasets includes COCO[30], LaSOT[13], GOT-10k[22], and TrackingNet[34]. Data augmentation techniques include Random horizontal flip and brightness jittering.

**Inference**. The size of GR memory $N_{max}$ is set to $3 \times N_z$ by default. The update interval of the GR memory is set to 5 for $t <= 100$, doubled every 100 frames until $t = 500$, and then remains 160. The gradual increase in the update interval is to reduce the accumulation of model errors caused by the template from inaccurate tracking results. 3 stages relevance attention is set to $floor(3 \times N_z \times 0.9)$, $floor(3 \times N_z \times 0.8)$ and $floor(3 \times N_z \times 0.7)$ respectively, where $floor$ means rounding down to the nearest integer. We employ a single NVIDIA GeForce RTX 3090 for inference.

### 4.2 State-of-the-Art Comparisons

We compare the proposed RFGM with state-of-the-art methods on five tracking benchmarks, including large-scale benchmarks(TrackingNet[34], GOT-10k[22], and LaSOT[13]) and two small-scale benchmarks(OTB[45] and UAV123[33]).

**TrackingNet**. TrackingNet is a dataset containing large-scale testing videos, with a total of 511 testing videos. As reported in Tab.1, RFGM-B256 performs better than the recent state-of-the-art method SimTrack[4] with 1.4 %AUC score improvement. Importantly, we only use ViT-B as our encoder and SimTrack use ViT-L as the encoder. In addition, our model significantly outperforms other models with search resolution smaller than 300, achieving an AUC of 84.7%, $P_{norm}$ of 89.6 %

Table 2: Comparisons with state-of-the-art methods with search resolution $> 300 \times 300$ on three large-scale benchmarks. The top three metrics are highlighted with **red**, blue and green fonts.

| | Method | TrackingNet [34] | | | GOT-10k* [22] | | | LaSOT [13] | | |
|---|---|---|---|---|---|---|---|---|---|---|
| | | AUC | $P_{Norm}$ | P | AO | $SR_{0.5}$ | $SR_{0.75}$ | AUC | $P_{Norm}$ | P |
| | **RFGM-B256** | **84.7** | **89.6** | **83.6** | **74.1** | **84.6** | **71.8** | 70.3 | **82.0** | **76.4** |
| Resolution$> 300 \times 300$ | OSTrack-384 [54] | 83.9 | 88.5 | 83.2 | 73.7 | 83.2 | 70.8 | 71.1 | 81.1 | **77.6** |
| | SwinTrack-384 [28] | 84.0 | - | 82.8 | 72.4 | 80.5 | 67.8 | **71.3** | - | 76.5 |
| | Mixformer-L [9] | 83.9 | 88.9 | 83.1 | - | - | - | 70.1 | 79.9 | 76.3 |
| | Mixformer-22k [9] | 83.1 | 88.1 | 81.6 | 70.7 | 80.0 | 67.8 | 69.2 | 78.7 | 74.7 |
| | AiATrack [15] | 82.7 | 87.8 | 80.4 | 69.6 | 80.0 | 63.2 | 69.0 | 79.4 | 73.8 |
| | UTT [31] | 79.7 | - | 77.0 | 67.2 | 76.3 | 60.5 | 64.6 | - | 67.2 |
| | CSWinTT [41] | 81.9 | 86.7 | 79.5 | 69.4 | 78.9 | 65.4 | 66.2 | 75.2 | 70.9 |
| | STARK [51] | 82.0 | 86.9 | - | 68.8 | 78.1 | 64.1 | 67.1 | 77.0 | - |
| | RTS [37] | 81.6 | 86.0 | 79.4 | - | - | - | 69.7 | 76.2 | 73.7 |
| | Unicorn [50] | 83.0 | 86.4 | 82.2 | - | - | - | 68.5 | - | - |

Table 3: Comparisons with state-of-the-art methods on two small-scale benchmarks. The top three metrics are highlighted with **red**, blue and green fonts.

| | SiamRPN++ [26] | PrDiMP [11] | TransT [5] | TrDiMP [43] | STARK [51] | AiATrack [15] | CSWinTT [41] | ToMP [32] | Mixformer-L [9] | ours |
|---|---|---|---|---|---|---|---|---|---|---|
| OTB | 69.6 | 69.6 | 69.4 | 71.1 | 68.5 | 69.6 | - | 70.1 | 70.4 | **71.5** |
| UAV123 | 61.3 | 68.0 | 69.1 | 67.5 | 69.1 | **70.6** | 70.5 | 66.9 | 68.7 | 68.5 |
| Speed(fps) | 35 | 30 | 50 | 35 | 42 | 38 | 12 | 24 | 18 | 71 |

and P of 83.6. Notably, as shown in Tab.2 comparisons, our model also surpasses all methods with search resolution greater than 300 in the TrackingNet benchmark.

**GOT-10k**. GOT-10k consists of a training set of 10,000 videos and a validation set of 180 videos. Moreover, the training and test sets do not overlap, requiring trackers to have a strong generalization ability for unseen data. Following the official evaluation rules, we only evaluate models trained on the GOT-10k training set. As reported in Tab.1 and Tab.2, similar to the results on TrackingNet, our method surpasses all methods with resolutions greater than 300 and smaller than 300 by using only a search resolution of 256, achieving a 74.1% of AO, 84.6%of SR0.5, and 71.8% of SR0.75. This demonstrates the strong generalization ability of our model to unseen data.

**LaSOT**. LaSOT is a comprehensive benchmark for long-term tracking tasks, including a test set comprising 280 videos. The average length of the videos in the test set is 2448 frames. As shown in Tab.1, our model achieves the highest performance in $P_{norm}$ and P, with scores of 82.0% and 76.4% respectively, outperforming other methods. It only lags behind SimTrack by 0.2% in terms of AUC, mainly due to their use of ViT-L while we only use ViT-B. In comparison with methods using search resolutions greater than 300 in Tab.2, our model still demonstrates competitive performance on the Lasot dataset.

**OTB**. As shown in Tab.3, OTB is a classic dataset in the field of object tracking, consisting of 100 video sequences. RFGM demonstrates excellent performance on this dataset, This demonstrates that our method also exhibits generalization performance on the classical object tracking dataset.

**UAV123**. As shown in Tab.3, UAV123 is a dataset specifically designed for UAV (Unmanned Aerial Vehicle) target tracking, consisting of 123 video sequences with relatively small objects. RFGM achieves competitive performance on the dataset. Achieving a frame rate of 71 FPS, our approach significantly outperforms other methods in terms of speed.

### 4.3 Ablation study

**Why does GR memory work?** We conduct ablation experiments on different memory types, as shown in Tab.4. When using only one template, the performance is low because there is no template update to adapt to the appearance variations of the target. When using regular memory updating with a fixed interval, competitive performance is achieved on GOT-10k and TrackingNet, but the AUC on LaSOT is only 64.5. This is because LaSOT is a long-term benchmarks, and errors can accumulate in the memory when updating the template, resulting in performance degradation. By using the center score from the prediction head, errors in the memory can be reduced, but it cannot store the

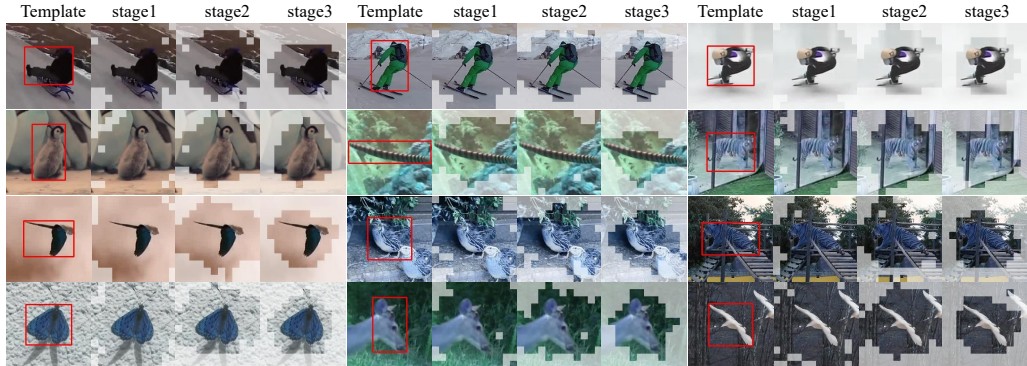

Figure 4: Visualization of relevance attention. Taking one template as an example, white areas represent discarded regions, while the remaining areas represent the regions selected by relevance attention. Stages 1 to 3 indicate the progressive application of three relevance attention layers.

Table 4: Ablation study on our proposed GR memory. One template represents only the use of the first frame as the template. Memory represents replacing an old template with a new template in a fixed interval. Score memory represents replacing the template according to the center score of the prediction head. The best metrics are highlighted with red fonts.

| Method | TrackingNet [34] | | | GOT-10k* [22] | | | LaSOT [13] | | |
|---|---|---|---|---|---|---|---|---|---|
| | AUC | $P_{Norm}$ | P | AO | $SR_{0.5}$ | $SR_{0.75}$ | AUC | $P_{Norm}$ | P |
| One template | 84.0 | 88.2 | 82.5 | 70.4 | 80.0 | 66.3 | 69 | 80.1 | 74.0 |
| Memory | 84.5 | 88.8 | 82.65 | 71.9 | 82.0 | 69.4 | 64.5 | 73.9 | 68.2 |
| Score memory | 84.6 | 88.9 | 82.7 | 74.3 | 84.0 | 70.8 | 69.7 | 81.5 | 76.2 |
| GR memory | 84.7 | 89.6 | 83.6 | 74.1 | 84.6 | 71.8 | 70.3 | 82.0 | 76.4 |

most representative features. In contrast, our proposed GR memory can store the most representative features throughout the entire video sequence and also reduce error accumulation in the memory, resulting in the highest performance. The visualization of GR memory updates is illustrated in Fig5.

**Attention comparison**. As reported in Tab.5, using only the attention mechanism from ViT cannot adaptively select the most suitable features from the reference for the current search frame. Therefore, its performance is lower than our proposed relevance attention. It is worth noting that the baseline here includes GR memory, where all features have already undergone a round of filtering using relevance attention within the GR memory. In other words, even within the filtered GR memory,

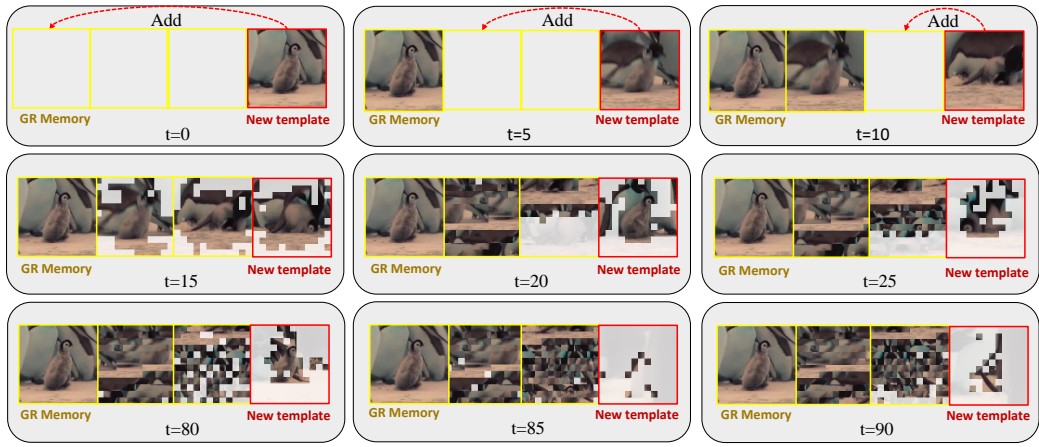

Figure 5: Visualization of GR memory updates. Over time t, the GR memory accumulates an increasing number of representative penguin features. In the second and third rows, white tokens represent discards, while the other tokens are retained in memory.

Table 5: Attention represents to use the original multi-head attention of ViT[12] in encoder. Relevance attention represents using relevance attention in our encoder. The best metrics are highlighted with red fonts.

| Method | TrackingNet [34] | | | GOT-10k* [22] | | | LaSOT [13] | | | Macs | Parameter |
|---|---|---|---|---|---|---|---|---|---|---|---|
| | AUC | $P_{Norm}$ | P | AO | $SR_{0.5}$ | $SR_{0.75}$ | AUC | $P_{Norm}$ | P | | |
| Attention | 84.5 | 89.3 | 83.7 | 73.9 | 84.1 | 72.6 | 69.8 | 81.3 | 75.7 | 40.00G | 92.12M |
| Relevance attention | 84.7 | 89.6 | 83.6 | 74.1 | 84.6 | 71.8 | 70.3 | 82.0 | 76.4 | 37.66G | 92.35M |

Table 6: Ablation experiments on memory size. The numbers 64, 192, 256, and 448 represent the token number in the memory

| | 64 | 192 | 256 | 448 |
|---|---|---|---|---|
| AUC | 69.0 | 70.3 | 69.6 | 69.5 |
| $P_{norm}$ | 80.1 | 82.0 | 81.5 | 81.3 |
| P | 74.0 | 76.4 | 75.6 | 75.6 |

Table 7: The MACs and parameters are measured with and without deformation attention

| | w/o | 1 stage | 2 stages | 3 stages |
|---|---|---|---|---|
| Macs | 40.00G | 39.63G | 38.85G | 37.66G |
| Parameters | 92.12M | 92.20M | 92.27M | 92.35M |

further performance improvement can be achieved by applying relevance attention. Additionally, relevance attention can reduce the number of computational parameters (40.00G VS 37.66G) while maintaining negligible parameter overhead(92.12M VS 92.35M). Tab.7 also presents the macs and parameters with or without relevance attention. The visualization of relevance attention is shown in Fig.4.

**What is the best size of GR memory?** As shown in Tab. 6, we conducted ablation experiments on the size of the memory. We found that the larger memory size does not lead to better performance. In fact, when the memory size exceeds 192 tokens, the performance starts to decline. This is because storing templates in the memory based on incorrect tracking results can lead to error accumulation. With a larger memory size, more errors are accumulated as the tracking progresses. Therefore, the optimal memory size for RFGM is 192 tokens.

## 5    Conlusion and Limitation

In this paper, we present a novel tracking framework tracking(RFGM), consisting of relevance attention and GR memory. Relevance attention adaptively selects the most suitable features for the current search region, allowing for adaptation to target appearance variations and environmental changes. Additionally, we construct a GR memory that utilizes relevance attention to select features from incoming templates. Through multiple rounds of updates, the GR memory stores the most representative features from the entire video sequence. Comprehensive experiments demonstrate that our method achieves state-of-the-art performance, and ablation experiments verify the effectiveness of the proposed GR memory and relevance attention. Due to the template updates, incorrect information will be stored into the memory, resulting in the accumulation of distractors over the process of the object tracking, we plan to further investigate methods to reduce the accumulation of errors in the memory.

**Acknowledgement** This work was supported by National Key RD Program of China (No.2020AAA0108301), National Natural Science Foundation of China (No.62072112 and No.62106051), Scientific and Technological innovation action plan of Shanghai Science and Technology Committee (No.22511102202), Fudan University Double First-class Construction Fund (No.XM03211178), and the Shanghai Pujiang Program (No.21PJ1400600).

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
