# OpenReview forum: "Reading Relevant Feature from Global Representation Memory for Visual Object Tracking"
_NeurIPS.cc/2023/Conference — NeurIPS 2023 poster_

### Official Review · Reviewer_UhjK · 2023-07-03

**Soundness:** 3 good
**Presentation:** 3 good
**Contribution:** 3 good
**Rating:** 6
**Confidence:** 4

**Summary:**

In this paper the authors present a single object tracking algorithm that flexibly explores historical information and spatial information, in the terms of deformable memory and deformable attention, respectively. The proposed algorithm was tested on five standard benchmarks and performs better or similar to state-of-the-arts.

**Strengths:**

* The overall architecture of the proposed solution is solid and well designed.
* Adaptively selecting templates from past frames is an effective strategy and the proposed deformable memory solution seems a novel solution.
* Similarly, the deformable attention for selecting token/patches across spatial-temporal dimensions is novel, and nicely addressed the template update issue.
* The paper in the high level is well organized and presented.

**Weaknesses:**

* The title “deformable tracking” is misleading since it has been used for “tracking deformable objects”, such as deformable object tracking, deformable surface tracking, etc.
* The experimental results, compared with state-of-the-arts, show only marginal improvements (e.g., around or less than 1% in many cases) or even worse (e.g., on LaSOT and UAV123). While it is argued that a smaller backbone (ViT-B) is used, it would be more convincing to provide results using larger ones (e.g., ViT-L).
* The specific technique presentation is very confusing and painful to follow, especially the notations such as z^p, p^z (note that p is a row vector), etc.
* (Minor issues) Typos: L292, “?.”

**Questions:**

* What are the performances on TrackingNet/GOT-10K/LaSOT that do not distinguish the search resolutions (i.e., combination of Table 1 and Table 2)?
* Since the token selection is across frames, is it possible sometimes all selected tokens will from very few frames? Would that hurt the tracking performance?


I have read the author’s rebuttal and I raised my final rating accordingly.

**Limitations:**

Yes.

---

> ### Author Rebuttal · Authors · 2023-08-09
>
> We want to express our gratitude for your support and valuable insights. It's reassuring to know that you recognize the originality and efficacy of our research. We would greatly appreciate your continued support in championing our work.
> #### **Q1.The title “deformble tracking” is misleading...**
> Thank you very much for your suggestions. We will proceed to modify the titles to avoid ambiguity in the future.
> #### **Q2.The experimental results, compared with the state-of-the-arts, show only marginal improvements....it would be more convincing to provide results using larger ones (e.g., ViT-L).**
>
> **GOT-10k, TrackingNet, LaSOT** Among all datasets, GOT-10k, LaSOT, and TrackingNet are significant datasets in the field of tracking and are considered large-scale datasets. Using ViT-B16, we achieved a 2.8% performance improvement over Swintrack on AO and a 2.7% improvement on SR0.5 in the GOT-10k dataset. Furthermore, we observed a 3.6% enhancement over Ostrack-256 on SR0.75. On the TrackingNet dataset, our method outperformed SimTrack-large by 1.3% in AUC, and demonstrated gains of 1.8% and 1.6% in Pnorm and P, respectively, when compared to Ostrack-256. In the LaSOT dataset, our approach exhibited a 2.3% improvement in Pnorm over SimTrack, and a 1.2% improvement over Swintrack.
>
> **ViT-base-384**.In response to your suggestion to enhance the persuasiveness of our study, we proceeded to train the  ViT-B-384 model. Our model has achieved further improvements on ViT-B-384, attaining state-of-the-art (SOTA) performance. **Thanks for your advice again, after adding ViT-B-384, we significantly surpass the previous method on GOT-10k, TrackingNet, and LaSOT**.
> |Method|temporal|TrackingNet| GOT-10k|Lasot|
> |:-:|:-:|:-:|:-:|:-:|
> |||AUC, PNorm, P|AO, SR0.5, SR0.75|AUC, PNorm, P|
> |OSTrack-256-temporal|√|83.8, 89.0, 82.8 |73.4, 83.3, 72.1 |68.5, 77.9, 74.8|
> |OSTrack-384-temporal|√|83.9, 89.4, 83.4 |73.2, 83.1, 71.4|71.1 81.25, 77.8|
> |SwinTrack-256|√|81.1, -, 78.4|71.3, 81.9, 64.5|67.2, -, 70.8|
> |SwinTrack-384|√|84.0, -, 82.8|72.4, 80.5, 67.8|71.3, -, 76.5|
> |Mixformer-L |√|83.9, 88.9, 83.1|-, -, -|70.1, 79.9, 76.3|
> |Mixformer-22k |√|83.1, 88.1, 81.6|70.7, 80.0, 67.8|69.2, 78.7, 74.7|
> |AiATrack|√|82.7, 87.8, 80.4|69.6, 80.0, 63.2|69.0, 79.4, 73.8|
> |STARK|√ |82.0, 86.9, -|68.8, 78.1, 64.1|67.1, 77.0, -|
> |SimTrack|√|83.4, 87.4, -|69.8, 78.8, 66.0|70.5, 79.7, -|
> |**DefTrack-256**|√|**84.7, 89.6, 83.6**|**74.1, 84.6, 71.8**|**70.3, 82.0, 76.4**|
> |**DefTrack-384**|√|**84.9, 89.6, 84.3**|**75.6, 85.6, 72.5**|**72.5, 84.5, 78.7**|
>
> **The newly introduced datasets: TNL2k, AvisT and NfS** Additionally, we introduced three new tracking datasets. Observing the experimental results, we significant surpass previous method by 3 to 7 % on AUC. we achieved state-of-the-art (SOTA) performance on seven out of eight datasets, providing substantiated evidence for the effectiveness of our exploration into historical frame information.
> |Method|temporal|Avist| TNL2k|NfS|
> |:-:|:-:|:-:|:-:|:-:|
> |||AUC, OP50, OP75|AUC, P|AUC|
> |SiamRPN++|x|39.0, 43.5, 21.2|41.3, 41.2|5| 50.2|
> |Ocean|x|38.9, 43.6, 20.5|38.4, 37.7|49.4|
> |OSTrack-256|x|54.2, 63.2, 42.2|54.3, 54.3|64.7|
> |OSTrack-384|x|57.7, 67.3, 48.3|55.9, 56.7|66.5
> |Mixformer|√|56.5, 66.3, 45.1|55.3, 55.3|64.9|
> |**DefTrack-256**|√|**59.0, 68.1, 49.2**|**61.3, 62.3**|**66.6**|
> |**DefTrack-384**|√|**62.2, 71.9, 53.5**|**62.4, 63.4**|**68.6**|
> #### **Q3.The specific technique presentation is .....Typos: L292,**
> We apologize for any confusion. The notation z^p represents the embedding obtained from the ViT after passing through convolutional layers, encompassing various tokens p^z. We will proceed to make adjustments to ensure clearer expression and rectify any typographical errors.
> #### **Q4.What are the performances on TrackingNet/GOT-10K/LaSOT that do not distinguish the search resolutions (i.e., combination of Table 1 and Table 2)?**
> Typically, higher resolutions lead to improved model performance, but larger resolutions also entail greater computational demands. To address this, we introduced results for the ViT-B-384 model, which exhibits enhanced performance, as indicated in the results.
> #### **Q5.Since the token selection is across frames, is it possible sometimes all selected tokens will from very few frames? Would that hurt the tracking performance?**
> We have visualized the updates to our memory, and visualization results of the deformable memory update process can be found in the supplementary materials submitted with our initial manuscript. From the visualization, interestingly, our deformable memory aggregates discriminative features from various historical frames, rather than relying solely on a small subset of frames.

---

> > ### Comment · Reviewer_UhjK · 2023-08-15
> >
> > The rebuttal addresses partly my concerns, so I would upgrade final rating accordingly.

---

> > > ### Author Response · Authors · 2023-08-16
> > >
> > > We are delighted that our rebuttal has addressed your concerns. We sincerely appreciate your recognition and the increased score for our work. Our research primarily addresses a longstanding and fundamental issue in the field of visual object tracking: changes in target appearance and background. We hope that our work, driven by a comprehensive exploration of temporal information, can provide valuable inspiration to the domain of visual object tracking.

---

### Official Review · Reviewer_nyT7 · 2023-07-06

**Soundness:** 3 good
**Presentation:** 3 good
**Contribution:** 2 fair
**Rating:** 5
**Confidence:** 4

**Summary:**

The paper presents a method to dynamically select templates for visual tracking. Based on the vision transformer (ViT) framework, template selection is formulated as token selection. A prediction module is proposed to rank tokens. Top-K tokens from the history memory is selected to interact with tokens from the search region. The proposed method is evaluated on five benchmark datasets and achieves very competitive results compared with state-of-the-art methods.

**Strengths:**

The paper present a method to dynamically select tokens from template memory which are used to interact with tokens from the search region. It can effectively select most relevant tokens and discard irrelevant tokens, therefore improving tracking performance.

The proposed method achieves very competitive performance compared with state-of-the-art methods. Also, ablation study is conducted to validate the effectiveness of different components in the proposed method. The proposed method runs fast at 71 FPS.

**Weaknesses:**

The novelty of the proposed method is limited. The idea of dynamic token selection has explored in previous work, e.g. [55], [A] and [B].
Especially, in both [A] and [B], the gumbel softmax is also used to learning the token score prediction module. The idea of token selection presented in this paper is  similar to the ideas in [A] and [B].
[A] Generalized Relation Modeling for Transformer Tracking, Gao et al, arxiv 2023
[B] Dynamicvit: Efficient vision transformers with dynamic token sparsification, Rao et al, Neurips 2021.

The comparison with state-of-the-art methods is not very fair. Most previous methods listed in Tables 1 and 2 only used a single template from the first frame, while the proposed method uses multiple templates. It is expected that using multiple templates can boost tracking performance. How is the performance of the proposed method with search region resolution of 384 x 384?

**Questions:**

See Weaknesses.

**Limitations:**

See Weaknesses.

---

> ### Author Rebuttal · Authors · 2023-08-09
>
> We highly value your constructive and insightful feedback. In terms of novelty, please allow me to reiterate the distinctive aspects of our research
> #### **Q1. The idea of dynamic token selection has explored in previous work...**
> ##### **Different Motivations:**
> GRM's motivation is the pollution of the template by the background of the search during cross-attention, thereby affecting the object localization in the search region. Therefore, GRM classifies tokens from the search features to alleviate the pollution of background features from the search for template features. DVIT, aims to enhance model speed by reducing the number of tokens in the search area.
>
> In contrast, our approach involves the cross-frame selection of the most discriminative features within a video sequence to construct a deformable memory, addressing the challenge of object tracking concerning changes in object appearance and background. Besides, we employ deformable attention to select the most relevant features from the memory with respect to the current search area. This enhances the tracker's robustness to object appearance changes.
>
> ##### **Different Target Focus:**
> GRM focuses on the classification of tokens within the search area, DVIT targets the discarding of tokens in the predicted image, whereas our method focus on the selection of tokens from historical frame and tokens within the memory.
>
> #### **Q2. the gumbel softmax is also .... the token score prediction module...**
> ##### **Our contribution does not lie in Gumbel-Softmax:**
>  As mentioned in our introduction. It serves merely as a technique employed to implement our proposed deformable tracking framework. Similar to using convolutions to construct a neural network or employing transformers for multi-modal interactions.
> ##### **Our Contribution:**
> Given that target deformation poses an inherent and highly challenging obstacle in object tracking, with no satisfactory resolution thus far, recent advancements in tracking performance have primarily stemmed from the utilization of powerful backbones and pretrained models. These advances, however, have largely disregarded an in-depth exploration of the fundamental problem itself.
>
> Prior research has either relied solely on single-template object tracking methods, which fail to address the challenges posed by variations in object appearance or has employed simplistic strategies such as adding or discarding templates for memory updates, as seen in methods like Mixformer, Aiatrack, Swintrack, and Stark. Such approaches lead to either an excessive number of templates stored in memory, significantly impeding tracker speed, or an inability to retain the most discriminative features from historical frames when utilizing a smaller memory. Moreover, certain features in the memory dynamically evolve over time and may no longer appear in the current search area, potentially affecting performance.
>
> In order to address these issues, we introduce deformable attention, which enables the retention of the most discriminative features (at the token level) across the entire video sequence, while eliminating undesired features to construct a robust memory. Simultaneously, deformable attention is leveraged to access template features most relevant to the current frame, thereby adapting to changes in target appearance. To the best of our knowledge, our work is the first to introduce this novel concept in both the realm of visual object tracking (VOT).
> #### **Q3. The comparison with state-of-the-art methods is not very fair... it is expected that using multiple templates...**
> The high-performing trackers in Table 1 and Table 2 adopt a multi-template approach, such as Mixformer, Swintrack, Aiatrack, and Stark. In addition, based on your valuable comment, for fair comparisons, we conducted multi-template experiments on the high-performance single-template method OSTrack (baseline for DefTrack).
> While we observe performance gains for short video datasets like Got10k (averaging 100-150 frames), we do not witness similar improvements for longer video datasets such as TrackingNet (averaging 441.5 frames) and LaSOT (averaging 2448 frames). This is attributed to the increased challenges posed by longer videos with more complex and challenging variations in target appearance and background. Simply relying on multiple templates is insufficient to address this issue.
>
> Furthermore, we conducted experiments for the **DefTrack 384x384 model**. As shown in the tables, in comparison to multi-template methods, we have still achieved the highest performance. This outcome underscores the effectiveness of our approach in thoroughly harnessing the features from historical frames, rendering it more robust in handling variations in target appearance changes.
>
> | Method |temporal| TrackingNet| GOT-10k|Lasot|
> |:-:|:-:|:-:|:-:|:-:|
> |OSTrack-256|×|83.1, 87.8, 82.0 |71.0, 80.4, 68.2|69.1, 78.7, 75.2|
> |OSTrack-256-temporal |√|83.8, 89.0, 82.8|73.4, 83.3, 72.1 |68.5, 77.9, 74.8|
> |OSTrack-384 |×|83.9, 88.5, 83.2|73.7, 83.2, 70.8|71.1, 81.1, 77.6|
> |OSTrack-384-temporal |√|83.9, 89.4, 83.4|73.2, 83.1, 71.4|71.1 81.25, 77.8|
>
> |Method|temporal|TrackingNet| GOT-10k|Lasot|
> |:-:|:-:|:-:|:-:|:-:|
> |||AUC, PNorm, P|AO, SR0.5, SR0.75|AUC, PNorm, P|
> |OSTrack-256-temporal|√|83.8, 89.0, 82.8 |73.4, 83.3, 72.1 |68.5, 77.9, 74.8|
> |OSTrack-384-temporal|√|83.9, 89.4, 83.4 |73.2, 83.1, 71.4|71.1 81.25, 77.8|
> |SwinTrack-256|√|81.1, -, 78.4|71.3, 81.9, 64.5|67.2, -, 70.8|
> |SwinTrack-384|√|84.0, -, 82.8|72.4, 80.5, 67.8|71.3, -, 76.5|
> |Mixformer-L |√|83.9, 88.9, 83.1|-, -, -|70.1, 79.9, 76.3|
> |Mixformer-22k |√|83.1, 88.1, 81.6|70.7, 80.0, 67.8|69.2, 78.7, 74.7|
> |AiATrack|√|82.7, 87.8, 80.4|69.6, 80.0, 63.2|69.0, 79.4, 73.8|
> |STARK|√ |82.0, 86.9, -|68.8, 78.1, 64.1|67.1, 77.0, -|
> |**DefTrack-256**|√|**84.7, 89.6, 83.6**|**74.1, 84.6, 71.8**|**70.3, 82.0, 76.4**|
> |**DefTrack-384**|√|**84.9, 89.6, 84.3**|**75.6, 85.6, 72.5**|**72.5, 84.5, 78.7**|

---

> > ### Author Response · Authors · 2023-08-19
> > **The further supplements in the rebuttal serve to address the concerns raised by the reviewer.**
> >
> > #### **Q1. Reply to the reviewer's suggestions**
> > **I sincerely appreciate your valuable feedback, which has greatly contributed to improving our manuscript. Following your valuable suggestions, we have incorporated multiple templates into both OSTrack256 and OSTrack384 (our baseline), also trained DefTrack-B384x384 model, as demonstrated in the first table and the second of the above rebuttal.**
> >
> > This adjustment helps to better demonstrate that our method does not solely rely on multiple templates to enhance performance. Instead, it achieves high performance by constructing a global, cross-frame memory with discriminative features throughout the entire video sequence, and by selecting the most relevant features from this memory.
> >
> > #### **Q2. The challenges we aimed to address**
> > **1). Historical frame features are crucial for adapting to changes in target appearance and background, a long-standing and significant challenge in the field of visual object tracking. This challenge has yet to be effectively addressed.** Previous methods utilized all historical frame information within the memory, but this approach is intuitively flawed. For instance, if features present in the first and second frames no longer appear in the tenth frame, it could potentially harm tracking performance. Therefore, we propose selecting relevant features from memory frames to aid in adapting to changes in target appearance and background, addressing this fundamental challenge. This approach has enabled our tracker to reduce computational complexity and enhance tracking performance. Moreover, quantitative experimental results validate its effectiveness. **To the best of our knowledge, we are the first to introduce this concept in both the VOT(Visual Object Tracking) and VOS (Visual Object Segmentation) domains.**
> >
> > 2).Furthermore, the memory construction methods used in previous approaches often involved simplistic addition or discarding of entire templates. This approach could severely impact tracking speed if the memory becomes too large or fail to store a comprehensive historical frame record if it's too small. **Recognizing the shortcomings of these method**, we have adopted a global approach to construct a memory with discriminative features that span across frames throughout the entire video sequence. This enhancement improves the tracker's robustness in adapting to changes in target appearance and background while avoiding excessive compromise to tracking speed.
> >
> > #### **Q3. To further validate the robustness of our approach, we introduced three new tracking datasets: TNL2K, Avist, and NfS.**
> > **In recent years, tracker performance has primarily been improved through powerful pre-trained models such as ViT-MAE and CVT-22K. The use of single frames or simple template additions and discarding techniques has been employed to update memory, disregarding the exploration of historical frames. To further validate the superiority of our approach over previous methods, we introduce three new datasets: TNL2K**, a large-scale tracking dataset with 700 test videos; **Avist**, a recently released 2022 tracking dataset; and **NfS**, a high-frame-rate tracking dataset. As shown in the table below, our tracker's performance significantly outperforms previous methods on these newly introduced datasets, achieving AUC improvements ranging from 3% to 7%. This further underscores the effectiveness of the memory we have constructed and the selection of relevant features from this memory for video object tracking.
> > |Method|Temporal|Avist|TNL2k|NfS|
> > |:-:|:-:|:-:|:-:|:-:|
> > |||AUC, OP50, OP75|AUC, P|AUC|
> > |SiamRPN++|x|39.0, 43.5, 21.2|41.3, 41.2|50.2|
> > |Ocean|x|38.9, 43.6, 20.5|38.4, 37.7|49.4|
> > |OSTrack-256|x|54.2, 63.2, 42.2|54.3, 54.3|64.7|
> > |OSTrack-384|x|57.7, 67.3, 48.3|55.9, 56.7|66.5|
> > |STARK|√|51.1, 59.2, 39.1|-, -|65.2|
> > |ToMP|√|51.9 59.5 38.9|-, -|-|
> > |Mixformer|√|56.5, 66.3, 45.1|55.3, 55.3|64.9|
> > |**DefTrack-256**|√|**59.0, 68.1, 49.2**|**61.3, 62.3**|**66.6**|
> > |**DefTrack-384**|√|**62.2, 71.9, 53.5**|**62.4, 63.4**|**68.6**|
> >
> > **In summary, we would like to express our gratitude to the reviewer for providing valuable suggestions on our work. By incorporating supplementary experiments involving multiple templates with OSTrack and experiments with the DefTrack-B384X384 model, the quality of our manuscript has been further improved. The introduction of three new tracking datasets, TNL2K, Avist, and NfS, has additionally bolstered the validation of the effectiveness and robustness of our approach.**

---

> > ### Comment · Reviewer_nyT7 · 2023-08-20
> >
> > The rebuttal addresses most of my concerns. I would upgrade my rating accordingly. More discussion about dynamic token selection needs to be included, e.g. [A] and [B]. The implemetation of  OSTrack-256-temporal and OSTrack-384-temporal is a bit weak. More sophisticated temporal update stratigies could be considered. I also agree with Reviwer UhjK that the use of "deformble tracking" is a not very appropriate. The descriptions of the proposed method need to be modified accordingly.

---

> > > ### Author Response · Authors · 2023-08-20
> > >
> > > Dear Reviewer,
> > >
> > > We are delighted that our rebuttal has effectively addressed your concerns. We also appreciate your recognition of our work and the increased score. **Regarding your suggestion to include more update strategies for OSTrack, due to the impending closure of the discussion period, we are currently working under tight time constraints to train OSTrack models with different update strategies. However, we are committed to making every effort to develop OSTrack models with various update strategies and provide you with a response by August 21st at 1 pm EDT**.
> > >
> > > Once again, we sincerely thank you for your valuable feedback.
> > >
> > > Best regards

---

> > > ### Author Response · Authors · 2023-08-21
> > >
> > > Dear reviewer:
> > >
> > > **Q1. Although the time is tight, based on your valuable suggestions,  we have done experiments on different strategies of OSTrack, resolution of 256.**
> > >
> > > **Single-temp**: This is the baseline of OSrack using a single template.
> > >
> > > **Multi-temp-Direct**:This is a strategy with multiple template updates without any assessment of template quality.
> > >
> > > **Multi-temp-Thred**:This is to use OSTrack's confidence score to evaluate the template quality and thus selectively update template.
> > >
> > > **Multi-temp-GRM**:This is using the strategy in GRM for memory selection.
> > >
> > > **Multi-temp-DViT**:This is using the strategy in DViT for memory selection.
> > >
> > > |Method|temporal|TrackingNet| GOT-10k|Lasot|
> > > |:-:|:-:|:-:|:-:|:-:|
> > > |||AUC, PNorm, P|AO, SR0.5, SR0.75|AUC, PNorm, P|
> > > |Single-temp|x|83.1, 87.8, 82.0 |71.0, 80.4, 68.2|69.1, 78.7, 75.2|
> > > |Multi-temp-Direct|√|83.6,88.9,82.5|73.7,84.0,72.2|67.2, 76.2, 73.2|
> > > |Multi-temp-Thred|√|83.8, 89.0, 82.8 |73.4, 83.3, 72.1 |68.5, 77.9, 74.8|
> > > |Multi-temp-GRM|√|83.6, 89.3, 83.0|73.8,83.9,71.9|68.9, 78.5, 75.4|
> > > |Multi-temp-DViT|√|83.7, 89.4, 83.2|73.9,84.3,72.7|67.9, 77.2, 74.0|
> > > |**Ours**|√|**84.7, 89.6, 83.6**|**74.1, 84.6, 71.8**|**70.3, 82.0, 76.4**|
> > >
> > >
> > >
> > > **Analysis**:
> > >
> > > **GOT-10k**: From the above results, all multi-template strategies can be improved on the short-term dataset GOT-10k(averaging 100-150 frames). This is because there are fewer scenarios in which the target appearance and background change. Then in the actual application process, long-term tracking is more important than short-term tracking.
> > >
> > > **TrackingNet**: The improvement of other methods on TrackingNet(averaging 441.5 frames) is limited, no more than 1 %, and our method improves TrackingNet by 1.6 %, This is because TrackingNet is 4 times longer than the average GOT-10k video length.
> > >
> > > **LaSOT**: Importantly, on the long-term dataset LaSOT(averaging 2448 frames), except our method, the performance of other methods has dropped, especially Multi-temp-Direct and Multi-temp-DViT, because the long-term tracking dataset will face Challenges such as more frequent target and background changes increase the difficulty of tracking. Furthermore, Multi-temp-Direct has no way to evaluate the template quality, so it will introduce wrong templates. And Multi-temp-DViT is used for model acceleration, so there is a decline in performance. And Multi-temp-GRM can only be equal to Single-temp performance. And Multi-temp-Thred proves that the confidence evaluation of the tracker cannot cope with the challenge of long-term tracking of target and background changes in the scene.
> > >
> > >
> > >
> > > **Summary**: The above results demonstrate that our method can deal with the long-term challenge of target and background changes in visual object tracking. We have adopted a global approach to construct a memory with discriminative features that span across frames throughout the entire video sequence,  improving performance. We also select the relevant feature from memory to adapt to target and background changes.
> > >
> > > **Q2.Thanks for your valuable suggestion, we will re-name the memory and attention mechanism accordingly**
> > >
> > > 1). We propose a novel tracking framework that adapts to changes in target appearance and background by constructing a **global representation memory** at the token level across frames and reading from this memory to capture the most relevant features at the current time step.
> > >
> > > 2). We design a **dynamic relevance attention mechanism** for the search region to selectively extract template features from memory. Simultaneously, it is utilized to update the global representation memory at the token level, reduce memory consumption and enhance tracking speed.
> > >
> > > Best regards

---

### Official Review · Reviewer_jJzj · 2023-07-06

**Soundness:** 3 good
**Presentation:** 3 good
**Contribution:** 3 good
**Rating:** 7
**Confidence:** 3

**Summary:**

The authors propose an improved memory mechanism user-initialized tracking. Firstly, instead of storing a fixed subset of previous frame encodings in memory they store a subset of tokens that are dynamically selected at every iteration. This makes memory more efficient and increases discriminability. Secondly, they propose a ranking-based attention mechanism that only retrieves the most relevant tokens from memory. The later has only a marginal effect on the performance though. For some reason, they call both modules 'deformable' though what's deformable about them is not quite clear to me.

In an experimental evaluation on several user-initialized tacking benchmarks the proposed approach performs on par with the state-of-the-art while being significantly faster.

**Strengths:**

The proposed approach seems sound.

The advantages of the approach are convincingly demonstrated (sota performance with significantly higher FPS compared to strongest baselines).

The paper is relatively well written, though there are still quite a few typos and grammatical mistakes.

A minimal ablation study is reported.

**Weaknesses:**

Related work overview ignores the works in video objet segmentation (VOS), which is effectively the same problem as the one studied in this paper, just operating on masks, not boxes. As a results, sota approaches for both problems are very similar. In particular, XMem [Cheng and Schwing, ECCV'22] is highly relevant.

Presentation still requires significant improvement. In addition to the typos, sota results are incorrectly bolded for LaSOT large res (the proposed approach is not sota).

The benefits of the proposed approach are limited (though not insignificant).

**Questions:**

Please provide an overview of the related work in VOS and highlight the relationship between your contributions and latest VOS methods (in particular XMem).


**Limitations:**

Limitations are addressed in the manuscript.

---

> ### Author Rebuttal · Authors · 2023-08-10
>
> We extend our heartfelt gratitude for your perceptive insights and your recognition of the unique and meaningful contributions made by our research. Your support is highly valued, and we would be honored if you could serve as an advocate for our work.
>
> #### **Q1.Related works about VOS**
> **Related work about VOS**:
> We will include a review of VOS in our related work. VOS (Video Object Segmentation) is indeed closely related to VOT (Visual Object Tracking). I will incorporate VOS methods into the relevant literature later on. Specifically, VideoMatching achieves segmentation through matching, while MaskTrack propagates the mask from the previous frame to accommodate target appearance variations. OSVOS adapts to dynamic changes in target and background through online fine-tuning. FeelVOS and CFBI achieve high performance by matching the first frame and the previous frame. Subsequently, STM and STCN construct memory networks, enabling the model to store extensive historical frame information for robust object tracking and segmentation. This serves as ample evidence of the importance of historical frame information in VOS or VOT tasks. However, since memory incurs memory overhead and reduces model computation speed, it becomes challenging to store all historical frames during long-term object tracking and segmentation.
> Therefore, XMem introduces three distinct types of memory to facilitate long-term object tracking and segmentation. These include working memory, which stores recent frame-level information for immediate utilization; long-term memory, which stores features discarded from the working memory and compresses them to reduce memory overhead while preserving historical features to the greatest extent possible; and Sensor memory, which employs GRU to retain low-level information from the first few frames, compensating for the deficiencies of working memory and long-term memory in this aspect.
> #### **Differences with Xmem**
> **Different Motivations**: XMem is designed to reduce memory consumption for long-term tracking and segmentation while preserving a maximum amount of historical frame information. Over time, features that enter the memory first may be discarded. In contrast, our deformable memory is designed to selectively choose discriminative features across frames in a global manner, independent of time. Importantly, our deformable attention allows us to selectively retrieve features from memory that are most relevant to the current search region for tracking. For instance, certain features present in the first and second frames might no longer appear in the tenth frame, yet they remain useful for the current frame. In this context, we need to selectively read relevant features for tracking while minimizing the impact of irrelevant features on performance. XMem, on the other hand, utilizes all the information within memory. To the best of our knowledge, whether in VOT or VOS, we are the first to propose this selective feature retrieval concept from memory.
>
> **Different Memory Sizes**: XMem employs three memory types that are larger in size compared to our memory. Our memory, however, contains only 192 tokens, equivalent to the size of three templates.
>
> **Different Update Approaches**: XMem's working memory stores frame-level information. When the memory limit is reached, surplus features are transferred to the long-term memory, which employs a least-frequently-used (LFU) mechanism with a fixed threshold for feature discarding. The sensor memory is updated using GRU. In contrast, our approach relies entirely on deformable attention to select discriminative features as the memory from existing memory and new templates, discarding the remaining features without requiring a threshold.
>
> #### **Q2.Typos**
> We sincerely apologize for any typographical errors. We will make every effort to thoroughly review and rectify them in our subsequent revisions.
>
> #### **Q3.The benefits of the proposed approach are limited (though not insignificant).**
> **Memory ablation study**: The limited performance improvement observed in the previous memory ablation experiments can be attributed to the fact that I applied the deformable training approach to the regular memory methods as well. This led to a relatively constrained enhancement in results. However, upon removing the deformable training aspect, as illustrated in the table below, we showcase the efficacy of our deformable memory. Notably, there is a significant improvement, particularly evident in the LaSOT dataset where Pnorm exhibits a notable enhancement of 4.1%.
> |Method|Deformbale training|TrackingNet| GOT-10k|Lasot|
> |:-:|:-:|:-:|:-:|:-:|
> |||AUC, PNorm, P|AO, SR0.5, SR0.75|AUC, PNorm, P|
> |one template|x|83.7, 87.9, 82.2 |71.7, 81.4, 68.9|67.7, 76.7, 73.1|
> |Multi-template|x|83.8, 89.0, 82.8 |73.4, 83.3, 72.1|68.5, 77.9, 74.8|
> |**Deformable memory**|√|**84.7, 89.6, 83.6**|**74.1, 84.6, 71.8**|**70.3, 82.0, 76.4**|
>
> **Thank you once again for your valuable suggestions. The field of VOS has made significant progress in exploring the construction of memory and historical frames, as evidenced by contributions such as Xmem, STM, and STCN. In contrast, recent advancements in the tracking domain have primarily relied on powerful backbones and pre-trained models, neglecting the essential aspect of historical frame exploration. By incorporating VOS concepts into the related works, researchers in the tracking field can gain insights into the focal points of similar domains, thereby further enhancing the quality of our manuscript.**

---

> > ### Comment · Reviewer_jJzj · 2023-08-11
> > **Re:re**
> >
> > I thank the authors for their detailed response which addressed most of my concern. The only remaining issue is the claims of 'deformability' which are still not supported by anything in my opinion. If the authors agree to re-position the work to better reflect the actual contributions I will be happy to recommend the paper for acceptance.

---

> > > ### Author Response · Authors · 2023-08-12
> > >
> > > We sincerely appreciate the author's consideration of our rebuttal and recognition of our work. **Based on your valuable feedback, we are very willing to consider your suggestions to re-position our work in order to reflect our actual contributions**. After a thorough day of contemplation and discussion, we have redefined and re-positioned our work as follows:
> > >
> > > **1. Title**: Reading Relevant Features from Global Representation Memory for Video Object Tracking.
> > >
> > > **2. Our Contributions are now Re-summarized as follows**:
> > >
> > > 1). We propose a novel tracking framework that adapts to changes in target appearance and background by constructing a **global representation memory** at the token level across frames and reading from this memory to **capture the most relevant features** at the current time step.
> > >
> > > 2). We design a **dynamic relevance attention mechanism** for the search region to selectively extract template features from memory. Simultaneously, it is utilized to update the global representation memory at the token level, reduce memory consumption and enhance tracking speed.
> > >
> > > 3). We conduct systematic experiments and validate the effectiveness of the proposed designs. Our tracker achieves competitive performance on widely used benchmarks.
> > >
> > > **Thank you very much for your suggestions. With the work re-positioned, it can more directly reflect the contributions of this paper. I also believe that the ideas presented in the repositioned paper can provide certain inspirations to the field of visual object tracking. We are delighted to hear that you are willing to recommend the acceptance of our paper, and looking forward to receiving a score improvement from you.**.

---

> > > > ### Comment · Reviewer_jJzj · 2023-08-14
> > > > **Re:re**
> > > >
> > > > The authors' response has fully addressed my concerns and I am increase my final score to Accept.

---

> > > > > ### Author Response · Authors · 2023-08-15
> > > > >
> > > > > We greatly appreciate your recognition of our work and the increased score! Our approach, which addresses challenges posed by changes in target appearance and background through the exploration of historical frames, can provide insights into the field of visual object tracking. Combining the inherent challenges with powerful pre-trained models and backbones can further stimulate advancements in tracking.

---

### Official Review · Reviewer_vhAu · 2023-07-07

**Soundness:** 2 fair
**Presentation:** 3 good
**Contribution:** 2 fair
**Rating:** 4
**Confidence:** 5

**Summary:**

This work proposes a deformable memory for visual tracking task to mine historical target features for enhancing the tracking performance. It is achieved by accessing global cross-frame features. Additionally, it possesses the capability to retrieve pertinent historical information from the constructed memory, thereby reducing redundancy and mitigating the detrimental impacts of irrelevant background features. The effectiveness of the proposed method is extensively validated through experiments, demonstrating competitive performance.

**Strengths:**

1. Strong intuition to make use of historical features to enhance tracking performance.

2. Differentiable learning of the memory ranking with gumble softmax is interesting.

3. The design of an efficient online update in the deformable memory, which involves effective token discarding, is a noteworthy contribution.

**Weaknesses:**

1. The technical contribution of this work is limited.  The token ranking and memory discarding mechanism in this paper is proposed in previous work like OSTrack and STMTrack. The difference of detail design of this work is analogous to these components in previous works.

2. Unfair performance comparison, the effectiveness of this work is not verified.  Though strong performance gain is achieved in the proposed tracker according to Tab.1 and Tab.2, the compared tracker only utilize first frame template without temporal contexts. Authors use strong temporal context in both training and inference phase, the performance of the tracker is certainly stronger than spatial only trackers. This work should further compare the proposed work with spatial temporal trackers like START-ST and MixFormer-ST, and give ablation stuides about the performance variation about directly using two template concatentation and the deformable memory for mining temporal context to demonstrate the effectiveness of this work.

3. Reported tracking speed of 71 fps is not convincing.  According to tracking performance reported in Tab.3,  the proposed tracker is twice faster than ToMP and STARK. I think it impossible to have such tracking speed. Sec4.1 mentioned this work use ViT-B formulates a one-stream spatial temporal tracker, which has 12 attention layers. But as we known, ToMP has a much efficient network architecuture, why it's much more ineffecient compared to authors work ?

4. Writing of the paper need to be improved. The main figures in the paper is messy, especially Fig.2. and there are minor grammer errors, like line 144, vdeio -> video.

5. Advise on this work: The authors' attempt to address the long-standing and challenging problem of mining temporal context in visual tracking is indeed interesting. However, to further validate the effectiveness of their approach, it is recommended that they conduct a more fair comparison to existing methods. This will help establish the superiority of their proposed method more convincingly. Moreover, it would be beneficial for the authors to focus on developing techniques that effectively mine temporal context while minimizing the increase in the number of tokens in the one-stream tracking framework. This would help alleviate the computational overload brought about by switching from a spatial-only tracker to a spatial-temporal tracker. By addressing this challenge, the proposed approach would become more practical and efficient in visual tracking.






**Questions:**

plz refer to the weakness part.

**Limitations:**

nothing.

---

> ### Author Rebuttal · Authors · 2023-08-09
>
> We greatly appreciate your constructive and insightful comments. Thanks for your recognition of our work.
> Please allow me to restate the novelty of our work.
> #### **Q1.The token ranking is proposed in OSTrack**
> **Different Motivations**
> Ostrack employs attention weights to discard features by token ranking from the search region, thereby accelerating the model.
> Our method's motivation and contribution revolve around the exploration of the most fundamental challenge of objec appearance variation changes. The challenge has not yet been adequately addressed. Consequently, our approach leverages the capability of deformable attention to select features (ranking) in a global, cross-frame manner, constructing a memory with discriminative features to adapt to the core challenge of object appearance variation.
> Moreover,  we extract features most pertinent to the current search region from the memory as reference information. **To the best of our knowledge, we are the first to propose selecting historically relevant features most closely related to the current frame to address the challenges of object appearance and background changes**.
>
> **Targeting different Entities**
> Ostrack discards feature (token ranking) in the search region.
> In contrast, our approach involves selecting features (token ranking) from historical frame information and the memory.
>
> **Different Implementation Approaches**
> OSTrack employs attention weights for ranking, while we utilize Gumbel-Softmax to select tokens through a classification network, predicting scores implicitly.
>
> #### **Q2.The memory discarding mechanism is proposed in STMTrack**
> **Deformable Memory Superior to STMTrack Memory**
> STMTrack constructs a memory by a process of adding and discarding entire template to update it, which akin to Mixformer, AIATrack, and STARK. **However, we identified drawbacks in this memory construction method:** it fails to retain the most discriminative features from historical frames across frames. Moreover, methods like STMTrack require a considerable number of frames to adapt to changes in target appearance, significantly impeding tracker speed and demanding substantial gpu memory resources, resulting in a runtime of only 37fps.
>
> By leveraging the capabilities of deformable attention, we preserve the most discriminative features from historical frames across frames at the token level, creating a robust memory. This memory comprises only 192 tokens, enabling us to retain valuable information from historical frames. The smaller memory size reduces memory consumption, achieves 71fps, nearly twice as fast as STMTrack.
>
> **Template level VS Token level:** STMTrack adopts frame-by-frame updates, whereas our approach updates based on tokens, more efficient than STMTrack.
>
> #### **Q3. Unfair performance comparison...**
> Some high-performance trackers in Table 1 and Table 2 adopt a spatial-temporal approach, such as STARK, Mixformer, Swintrack, and AiATrack. Among these, STARK and Mixformer, as you mentioned, also employ a spatial-temporal strategy involving whole-frame updates. **Based on your valuable comment, for fair comparisons, we augmented the OSTrack with temporal context during both training and testing phases. As shown in the following table, we still achieved the highest performance compared to these spatial-temporal trackers.**
>  Because the utilization of simplistic template updates that involve discarding older frames and incorporating new ones results in limited feature storage, preventing the establishment of a holistic video representation.
> |Method|temporal|TrackingNet| GOT-10k|Lasot|
> |:-:|:-:|:-:|:-:|:-:|
> |||AUC, PNorm, P|AO, SR0.5, SR0.75|AUC, PNorm, P|
> |OSTrack-256-temporal|√|83.8, 89.0, 82.8 |73.4, 83.3, 72.1 |68.5, 77.9, 74.8|
> |OSTrack-384-temporal|√|83.9, 89.4, 83.4 |73.2, 83.1, 71.4|71.1 81.2, 77.8|
> |SwinTrack-256|√|81.1, -, 78.4|71.3, 81.9, 64.5|67.2, -, 70.8|
> |SwinTrack-384|√|84.0, -, 82.8|72.4, 80.5, 67.8|71.3, -, 76.5|
> |Mixformer-L |√|83.9, 88.9, 83.1|-, -, -|70.1, 79.9, 76.3|
> |Mixformer-22k |√|83.1, 88.1, 81.6|70.7, 80.0, 67.8|69.2, 78.7, 74.7|
> |AiATrack|√|82.7, 87.8, 80.4|69.6, 80.0, 63.2|69.0, 79.4, 73.8|
> |STARK|√ |82.0, 86.9, -|68.8, 78.1, 64.1|67.1, 77.0, -|
> |**DefTrack-256**|√|**84.7, 89.6, 83.6**|**74.1, 84.6, 71.8**|**70.3, 82.0, 76.4**|
> |**DefTrack-384**|√|**84.9, 89.6, 84.3**|**75.6, 85.6, 72.5**|**72.5, 84.5, 78.7**|
> #### **Q4.Ablation study on using two template concatentation and the deformable memory ...**
> We show one template, multi-template and our deformbale memory as the following table, which demonstrate the effectiveness of our proposed method.
> |Method|Deformbale training|TrackingNet| GOT-10k|Lasot|
> |:-:|:-:|:-:|:-:|:-:|
> |||AUC, PNorm, P|AO, SR0.5, SR0.75|AUC, PNorm, P|
> |one template|x|83.7, 87.9, 82.2 |71.7, 81.4, 68.9|67.7, 76.7, 73.1|
> |Multi-template|x|83.8, 89.0, 82.8 |73.4, 83.3, 72.1|68.5, 77.9, 74.8|
> |**Deformable memory**|√|**84.7, 89.6, 83.6**|**74.1, 84.6, 71.8**|**70.3, 82.0, 76.4**|
> #### **Q5.Reported tracking speed of 71 fps...**
> The speed of the model is influenced not only by its size but also by the resolution of the input images. STARK employs a resolution of 320 for input images, while TOMP uses 480. In contrast, our model resolution is 256. And they utilize a two-stream framework, leading to two rounds of feature extraction, which increases computational load. Furthermore, our model is built upon the one-stream framework OSTrack, which achieves a speed of 105.4 FPS using the ViT-B architecture.
> #### **Q6.There are minor grammar errors..**
> We apologize for any grammatical errors, and we will make careful revisions in subsequent versions.
>
> #### **Q7.Advise on minimizing the increase in the number of tokens in the one-stream tracking framework**
> Our deformable attention reduces the number of tokens in the template tokens every 3 layers, thereby reducing computational demands. In the future, we intend to delve deeper into research aimed at further reducing the token count.

---

> > ### Author Response · Authors · 2023-08-17
> > **The quantitative analysis of tracking speed serves to address the reviewer's concerns.**
> >
> > #### **1. The comparison of parameter count, FLOPs  and FPS among STARK, TOMP, OSTrack, and our proposed DefTrack is presented in the table below**:
> > |Method|Type|parameters|FLOPS|FPS|Device|
> > |----|:----:|:----:|----|----|----|
> > |Tomp50|two-stream|26.11M|25.71G|32.49fps|RTX 3090|
> > |Tomp101|two-stream|45.10M|44.22G|19.93fps|RTX 3090|
> > |STARK50|two-stream|23.35M|13.08G|37.43fps|RTX 3090|
> > |STARK101|two-stream|42.29M|22.68G|22.91fps|RTX 3090|
> > |OSTrack-B256|one-stream|92.12M|21.51G|109.62fps|RTX 3090|
> > |DefTrack-B256|one-stream|92.35M|37.66G|86.33fps|RTX 3090|
> >
> > The speed tests mentioned above were conducted after excluding data loading time, which helps prevent CPU reading bottlenecks and provides a more direct reflection of the model's speed on the GPU.
> >
> > As indicated in the table above, although utilizing Vit-B results in a higher parameter count or greater FLOPs compared to STARK, the one-stream-based methods such as OSTrack and our DefTrack significantly outpace Tomp and STARK in terms of speed, achieving 109 and 86 fps, respectively. OSTrack's speed of 109 fps is close to the reported 105.4 fps in their paper[**1**]. This represents the speed advantage of the one-stream approach over the two-stream approach.
> >
> > #### **2. We further analyzed why the one-stream approach is faster than the two-stream approach.**
> > **1) Network fragmentation reduces the degree of parallelism(As reported in ShuffleNetV2 [2]).**
> >
> > **Firstly**, “Multi-path” structure (fragmentation ) could decrease efficiency because it is unfriendly for devices with strong parallel
> > computing powers like GPU. It also introduces overheads such as kernel launching and synchronization[**2**]. STARK employs a two-stream approach, which is a multi-path structure. As shown in table below, STARK50 requires 6.56ms and 7.47ms for extracting template and search features.
> >
> > **Secondly**, to further validate this perspective, we conducted ablation experiments on the ViT model using both one-stream and two-stream approaches. As shown in the table below, when employing the two-stream approach, it takes 5.62ms and 6.19ms for extracting template and search features, totaling 11.81ms. In contrast, using the one-stream approach to simultaneously extract template and search features only requires 6.85ms, significantly less than 11.81ms.
> > |Method|Template backbone latency|Search backbone latency|Transformer latency|Device|
> > |----|:----:|:----:|:----:|:----:|
> > |STARK50|6.56ms|7.47ms|12.08ms|RTX 3090|
> > |STARK101|13.81ms|14.90ms|11.94ms|RTX 3090|
> >
> >
> > |Method|Template backbone latency|Search backbone latency|Overall latency|Device|
> > |----|:----:|:----:|:----:|:----:|
> > |two-stream-ViT|5.62ms|6.19ms|11.81ms|RTX 3090|
> > |one-stream-ViT|-|-|6.85ms|RTX 3090|
> >
> > **2)  Equal channel width minimizes memory access cost (MAC), (As reported in ShuffleNetV2 [2])**
> >
> > **Firstly**, for a convolutional or linear layer, having an equal number of input and output channels results in a lower memory access cost (MAC), which is conducive to improving the model's inference speed. An experiment[**2**]was conducted with ShuffleNetV2, wherein a network comprising 10 blocks was constructed, with each block consisting of two convolutional layers.  The first contains c1 input channels and c2 output channels, and the second is otherwise. As shown in the two tables[**2**] below, both on GPU and ARM platforms, the model achieves the fastest inference speed when the number of channels is equal.
> >
> > |c1:c2|(c1,c2)|GPU speed(Images/second)|
> > |----|:----:|:----:|
> > |1:1|(128,128)|1780|
> > |1:2|(90, 180)|1296|
> > |1:6|(52, 312)|876|
> > |1:12|(36, 432)|748|
> >
> > |c1:c2|(c1,c2)|ARM speed(Images/second)|
> > |----|:----:|:----:|
> > |1:1|(32,32)|76.2|
> > |1:2|(22, 44)|72.9|
> > |1:6|(13, 78)|69.1|
> > |1:12|(9, 108)|57.6|
> >
> > **Secondly**,s shown in the table below, transformer layers of STARK consist of 12 layers of transformers with an input of 256 channels and an output of 2048 channels, taking around 12ms. Adopting ViT's transformer, although also comprising 12 layers, maintains a equal input and output channel of 768 channels, requiring only 6.85ms.
> >
> > |Method|layer num|c1:c2|(c1,c2)|latency|
> > |-|:-:|-|:-:|:-:|
> > |STARK-transformer layers|12|1:8|(256,2048)|12.08ms|
> > |ViT-transformer layers|12|1:1|(768, 768)|6.85ms|
> >
> > **3. Summary**
> >
> > **The two-stream approach, being a multi-path structure, requires separate feature extraction for both template and search, which can hinder GPU parallelism. Additionally, the input-output channel ratio of 1:8 (256, 2048) in STARK's transformer layers impacts MAC. Therefore, based on the one-stream framework, our proposed DefTrack can indeed achieve high-speed tracking.**
> >
> >
> > [1] Ye B, Chang H, Ma B, et al. Joint feature learning and relation modeling for tracking: A one-stream framework[C]//European Conference on Computer Vision(ECCV). 2022.
> >
> > [2] Ma N, Zhang X, Zheng H T, et al. Shufflenet v2: Practical guidelines for efficient cnn architecture design[C]//Proceedings of the European conference on computer vision (ECCV). 2018.

---

> ### Author Response · Authors · 2023-08-20
> **Dear Reviewer: we have provided a comprehensive response to your concerns through detailed explanations and thorough experimental validations.**
>
> Dear Reviewer:
>
> As the author-reviewer discussion period is coming to a close, we have provided a comprehensive response to your concerns through detailed explanations and thorough experimental validations. Could we kindly know if the responses have addressed your concerns?
>
> Thank you for your time and consideration.
>
> Best regards

---

### Author Rebuttal · Authors · 2023-08-10

We sincerely appreciate the thorough review provided by all the reviewers. The valuable feedback from the reviewers has significantly contributed to enhancing the quality of our manuscript. We extend our gratitude to Reviewer **jJzj** and Reviewer **UhjK** for acknowledging the novelty of our work. Their positive recognition of the innovation in our research is greatly appreciated. Furthermore, we kindly request Reviewer **vhAu** and Reviewer **nyT7** to reconsider our work after reviewing our response. Your reconsideration will be highly valued.

Based on the comments from the reviewers, I have summarized the strengths of our paper as follows:

1. Strong intuition to make use of historical features to enhance tracking performance.

2. The design of an efficient online update in the deformable memory, which involves effective token discarding, is a noteworthy contribution

3. The advantages of the approach are convincingly demonstrated (sota performance with significantly higher FPS compared to strongest baselines.

4. The paper presents a method to dynamically select tokens from template memory that are used to interact with tokens from the search region. It can effectively select the most relevant tokens and discard irrelevant tokens, therefore improving tracking performance. The proposed method achieves very competitive performance compared with state-of-the-art methods. Also, an ablation study is conducted to validate the effectiveness of different components in the proposed method. The proposed method runs fast at 71 FPS.

We have summarized our novelty as follows:

1. Adapting to variations in target appearance and background has consistently been the most fundamental and challenging aspect of object tracking, yet it has remained inadequately addressed. To tackle this issue, we propose the utilization of deformable attention to establish a robust memory that spans across frames, enabling global context awareness. Through token-level updates, our deformable memory encapsulates the most discriminative features from the entire video sequence, consequently enhancing tracking performance. The maintenance of token-level features contributes to reduced computational load, enabling our tracker to achieve a remarkable speed of 71 FPS.

2. We devised a deformable attention mechanism to filter out features irrelevant to the current temporal search region, enabling the selection of the most pertinent features for tracking at the present time step. To the best of our knowledge, we are the first to introduce this concept in VOT.

3. Our tracker demonstrates high performance across eight datasets, including the newly added Avist, TNL2K, and NFS. These datasets encompass GOT-10k, TrackingNet, LaSOT, AVisT, OTB, NFS, TNL2K, and UAV. Furthermore, our tracker achieves a notable speed of 71 FPS.

We hold the belief that these innovative contributions elevate the value and significance of our research in the realm of visual object tracking. We kindly request the reviewers to reassess our work considering these aspects and extend their support to our endeavor.

We intend to heed the insightful suggestions from the reviewers by incorporating additional essential experiments.  Additionally, a thorough review of the manuscript will be conducted to rectify any typographical and grammatical errors.

We have also provided comprehensive responses to each reviewer, meticulously addressing all specific points raised. We extend our thanks for the valuable feedback provided by all reviewers, as well as the dedication of the program chair and area chair. Your support in our endeavors would be greatly appreciated. We are try our best in our commitment to address the raised concerns and refining our manuscript accordingly.

---

### Decision · Program_Chairs · 2023-09-21

**Decision:**

Accept (poster)

**Comment:**

The submitted paper introduces a deformable memory for the visual tracking task, leveraging historical target features. Reviewers highlighted the strong intuition, effective token discarding mechanism, and significant FPS advantage as strengths. Concerns raised include possible limited novelty, performance comparisons, and some presentation issues. The authors, in their rebuttal, have addressed these concerns, emphasizing the uniqueness of their approach for visual tracking, and their competitive performance across multiple datasets. Given the positive aspects and potential impact, with a commitment from authors to address the feedback, this paper is recommended for acceptance.